# Uncertainty-Aware Learning for Zero-Shot Semantic Segmentation

**Ping Hu**[1]        **Stan Sclaroff**[1]        **Kate Saenko**[1,2]

[1]Boston University    [2]MIT-IBM Watson AI Lab

## Abstract

Zero-shot semantic segmentation (ZSS) aims to classify pixels of novel classes without training examples available. Recently, most ZSS methods focus on learning the visual-semantic correspondence to transfer knowledge from seen classes to unseen classes at the pixel level. Yet, few works study the adverse effects caused by the noisy and outlying training samples of the seen classes. In this paper, we identify this challenge and address it with a novel framework that learns to discriminate noisy samples based on Bayesian uncertainty estimation. Specifically, we model the network outputs with Gaussian and Laplacian distributions, with the variances accounting for the observation noise and uncertainty of input samples. Learning objectives are then derived with the estimated variances playing as adaptive attenuation for individual samples in training. Consequently, our model learns more attentively from representative samples of seen classes while suffering less from noisy and outlying ones, thus providing better reliability and generalization toward unseen categories. We demonstrate the effectiveness of our framework through comprehensive experiments on multiple challenging benchmarks, and show that our method achieves significant accuracy improvement over previous approaches for large-scale open-set segmentation.

## 1   Introduction

Semantic image segmentation aims to recognize and group pixels of the same object or stuff classes into segments [4, 16, 42, 64]. As a fundamental problem in computer vision, this task has attracted a lot of attention from the research community and achieved great success along with the development of deep learning in recent years [7, 8, 14, 15, 19, 24, 38, 45, 50, 51, 57, 59, 60, 63, 66]. Most of the existing methods focus on addressing the task over small and close sets of class labels, which relies on a large amount of training data to achieve effectiveness. Yet, due to the varying frequency of different object and stuff categories in natural scenes, annotations and samples for some categories may be difficult to acquire [46], thus posing the challenge in extending those conventional models to address large and open sets of categories.

To achieve effective zero-shot semantic segmentation (ZSS), existing efforts have been made [2, 28, 31, 44, 55, 62] by treating each pixel as an independent classification problem. And the classic zero-shot image recognition techniques [1, 3, 6, 18, 29, 35, 47, 52, 54, 61, 65] are directly applied to learn from seen classes the visual-semantic mappings, which are then transferred to unseen ones. Though achieving promising results, these methods may still suffer from several limitations. At first, these methods learn from pixels independently. Yet in images, category-consistent regions are more semantically meaningful than individual pixels. Thus learning with global information benefits the effectiveness of learned visual-semantic mappings. Besides, a more critical challenge is that most of the existing methods ignore the noisy and outlying samples of seen classes, which may cause adverse learning effects. As shown in Fig. 1 (a), closed-set learning with visual examples available for all the classes typically results in balanced visual-semantic mappings. However in zero-shot learning,

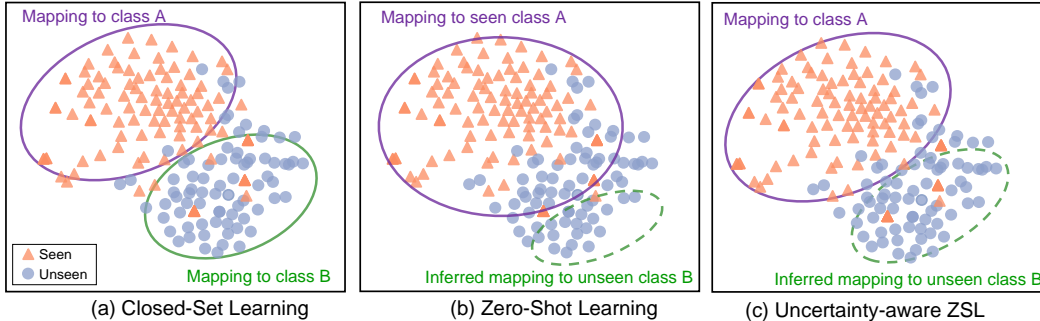

| (a) Closed-Set Learning | (b) Zero-Shot Learning | (c) Uncertainty-aware ZSL |

Figure 1: T-SNE visualization of visual feature encoded by ResNet [23]. Classes "A" and "B" are from PascalContext [42]. (a) Closed-set learning with "A" and "B". As both classes always have sufficient visual samples during training, the learnt visual-semantic mappings are balanced. (b) Zero-shot learning with seen class "A" and unseen class "B". Due to the lack of training samples for "B", the mapping learnt on "A" is sensitive to noisy and outlying samples, thus inferring biased mapping for the unseen class "B". (c) In uncertainty-aware zero-shot learning, noisy samples are attenuated during training. Thus visual-semantic mappings are learnt from representative samples of "A", and consequently infer a better mappings for unseen class "B".

without training data for unseen classes, the visual-semantic correspondence learned on seen classes is sensitive to the noisy and outlying samples. And consequently, as shown in Fig. 1 (b), sub-optimal mappings for unseen classes will be inferred due to the biased learning on seen classes.

To address these challenges of ZSS, in this paper we propose a novel framework that learns the visual-semantic mappings with global information, and leverages Bayesian uncertainty estimation [32, 37, 43] to automatically discriminate between representative samples and noisy ones during training. The proposed framework has two output branches, with one for pixel-wise prediction and the other learns to measure the overall segmentation quality. We model the output of each branch with a probabilistic distribution, by letting the network simultaneously estimate the mean and variance. The variance is related to the input sample's uncertainty [37, 43], thus allowing the model to explicitly account for the observation noise of the training data. Uncertainty-aware learning objectives are then derived with the estimated variances helping to adaptively strengthen representative training samples and attenuate noisy ones. Consequently, as illustrated in Fig. 1 (c) the model learns effective visual-semantic mappings from seen classes, which can be reliably transferred to unseen classes.

To the best of our knowledge, this is the first work that leverages uncertainty estimation based on Bayesian modeling to address noisy training samples in zero-shot learning tasks. Our main contributions are summarized as follows:

- We identify the problem of learning robust visual-semantic correspondence from noisy training samples in zero-shot learning tasks, and provide an effective solution based on data-dependent uncertainty estimation.

- We propose a novel deep probabilistic network for zero-shot semantic segmentation together with uncertainty aware losses that learn at image level and pixel level.

- We conduct extensive experiments on multiple benchmarks with large open-set classes, and show significant performance improvements over existing methods.

## 2 Related Work

**Zero-shot Semantic Segmentation.** Zero-shot learning is a highly active research area in computer vision and machine learning [21, 56]. Along with the recent advances in semantic segmentation [7, 8, 19, 25, 26, 38, 39, 53, 57, 59, 63], especially the fully convolutional network [39] that formulates the semantic segmentation tasks as a per-pixel classification problem, zero-shot semantic segmentation starts to attract attention from the community. Zhao *et al*. [62] formulate open-vocabulary scene parsing as a concept retrieval problem and utilize WordNet to build label relationships to segment large open-set classes in a hierarchical way. Xian *et al*. [55] propose a semantic projection network to unify both zero-shot and few-shot segmentation task. Bucher *et al*. [2] and Kato *et al*. [31] both apply variational visual-semantic mappings to adapt the semantic embeddings to the diverse conditions in the visual domain. Though achieving promising segmentation results, the aforementioned methods

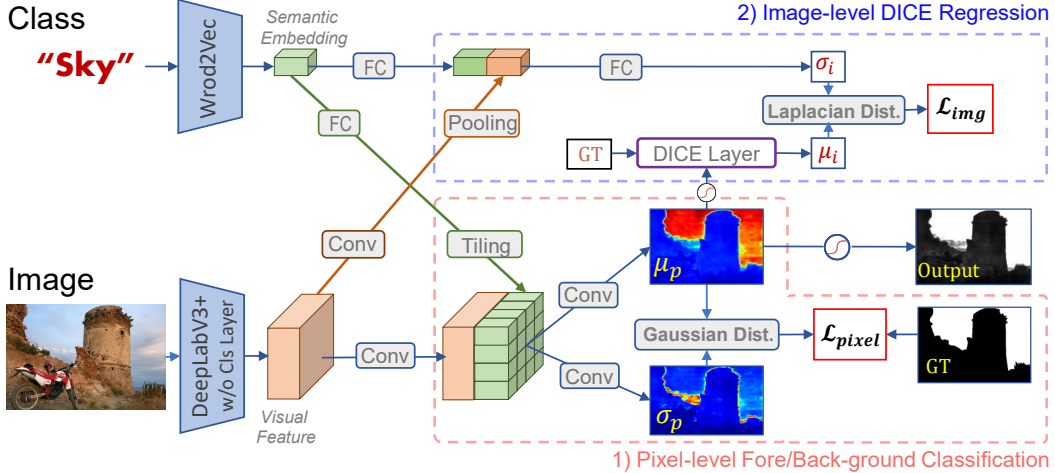

Figure 2: An overview of the proposed framework. A segmentation mask is generated from an image and a class tag. The model is trained on **seen** classes, and tested with **unseen** classes. **Training phase:** 1) In pixel-level learning, the network simultaneously estimates the mean $\mu_p$ and variance $\sigma_p$ to model the output as Gaussian distribution, and learning with an uncertainty-aware objective $\mathcal{L}_{pixel}$. 2) In image-level learning, we formulate the DICE loss as a Bayesian regression problem $\mathcal{L}_{img}$, and simultaneously estimating the mean $\mu_p$ and diversity $\sigma_p$ to parameterize a Laplacian distribution over DICE coefficient. **Testing phase:** Given an input image and a class label, the estimated mean of logits $\mu_p$ processed with a Sigmoid layer to be output.

often suffer from limitations like the neglect of segment-level segmentation quality, and the adverse learning effect caused by noisy training data. In contrast, we propose an effective framework with uncertainty-aware learning to address these challenges.

**Data-dependent Uncertainty Estimation.** Data-dependent uncertainty is also known as Heteroscedastic uncertainty [12, 20, 32] which models the observation noise of individual samples. It has been applied in deep Bayesian learning to help deep networks make reliable decisions and robust learning. Kendall *et al*. [32, 33] propose to account for the uncertainty with Gaussian distribution and learn to estimate the uncertainty for robust scene understanding. Ilg *et al*. [27] model uncertainty with a multiple hypothesis network for optical flow. Both Feng *et al*. [17] and Choi *et al*. [9] extend object detection frameworks to be probabilistic and learn to estimate the uncertainty of detection results. Khan *et al*. [34] aim at balanced learning from training data with both model and data uncertainty based on Monte Carlo sampling, which however incurs high computational costs. Levas *et al*. [36] extend the dropout variational inference with temperature scaling to calibrate model uncertainty. Besides serving as prediction confidence as in the above methods, uncertainty is also modeled in distributional feature representation for applications like person ReID [58] and facial recognition [5, 49] to achieve robust feature learning. Compared to these existing methods that all focus on closed-set tasks, our work extends the Bayesian uncertainty estimation to a new and challenging task of zero-shot semantic segmentation and presents an effective framework together with derived uncertainty-aware objectives.

## 3 Uncertainty-Aware Zero-Shot Semantic Segmentation

### 3.1 Zero-Shot Semantic Segmentation

In the scenario of zero-shot segmentation (ZSS), we aim to transfer the pixel-level visual-semantic mappings from seen classes to unseen ones. For simplicity, we formulate the task as an image segmentation problem conditioned on given semantic concepts. Formally, we split the category space into disjoint seen class set $\mathcal{S}$ and unseen class set $\mathcal{U}$. And consequently, the semantic embedding space $\mathcal{Y}$ for all the categories is also divided into two parts $\mathcal{Y}^s$ and $\mathcal{Y}^u$. We assume a training dataset with $n_s$ samples from the seen classes $\mathcal{S}$ is denoted as $\mathcal{D}^s = \{(x_i, y_i, z_i)\}_{i=1}^{n_s}$, where $x_i$ is an image in the image space $\mathcal{X}$, $y_i$ represents an embedding for a seen class in $\mathcal{Y}^s$, and $z_i$ is the corresponding mask in the binary segmentation space $\mathcal{Z}$. The goal for zero-shot semantic segmentation is to learn from $\mathcal{D}^s$ the conditional image segmentation model:

$$f_{zss}: \quad \{\mathcal{X}, \mathcal{Y}\} \to \mathcal{Z} \tag{1}$$

where $\mathcal{Y} = \mathcal{Y}^s \cup \mathcal{Y}^u$ meaning that the model doesn't only segment seen classes, but also need to be generalized to unseen classes.

## 3.2 Framework

The overview of our proposed uncertainty based learning framework for zero-shot semantic segmentation is illustrated in Fig. 2. To learn reliably and effectively from seen classes, we formulate uncertainty-aware learning at both pixel level and image level. Given an image and a target class, we first extract visual feature maps and semantic embedding vectors. Upon these, we apply concatenation operation and learn to estimate visual-semantic correspondence. In the training phase, we optimize the model with uncertainty at pixel level with stochastic Binary Cross Entropy (BCE) for the pixel-wise logits, and at image level with Bayesian regression for the DICE coefficient [41], so as to explicitly account for the noise and outlying samples in the training set. The two branches in the network are optimized simultaneously during training,

$$\mathcal{L} = \mathcal{L}_{img} + \lambda \cdot \mathcal{L}_{pixel} \tag{2}$$

where $\lambda$ is a weight, and $\mathcal{L}_{img}$ and $\mathcal{L}_{pixel}$ are the pixel level and image level uncertainty-aware losses that will be introduced in the following subsections.

## 3.3 Pixel-level Learning with Stochastic Binary Cross Entropy

With the help of FCN [39], deep CNN based semantic segmentation is efficiently converted into pixel-wise classification tasks. Given a training image, models typically learn equally from all the annotated pixels. However, among these there may exist two types of uncertain pixels: 1) atypical samples, whose features are less discriminative, e.g. pixels near boundaries; 2) label noise, which is caused by incorrect annotations. In the context of ZSS, learning from these atypical pixels may drive the model skew toward noisy and outlying samples, thus decreasing the generalization ability. In this section, we propose an uncertainty-aware method that learns robustly by estimating the noise level of pixels.

Given the visual feature maps and the semantic representation vector, we replicate the semantic vector to be the same spatial size as the feature map, and concatenate them along the channel dimension. Then, several $1 \times 1$ Conv layers are learned over the concatenated features to compare the visual-semantic relationship. Given $\mu_p$ to be the logit output by the network for a pixel, the segmentation probability is $p = Sigmoid(\mu_p)$, and upon which the Binary Cross Entropy loss is applied as,

$$\mathcal{L}_{bce}(p, z) = -z \cdot \log p - (1 - z) \cdot \log(1 - p) \tag{3}$$

where $z$ indicates the groundtruth label. This objective assumes the noise level is uniform through the sample space, which may lead to sub-optimal visual-semantic mappings.

Instead, we formulate the prediction to be probabilistic so as to account for noise of training data in a differentiable way. As shown in the lower branch of Fig. 2, in addition to the logits $\mu_p$ estimated on the concatenated features, an uncertainty parameter $\sigma_p$ is also estimated to quantify the noise level of the data. Then, we place a Gaussian parameterized with ($\mu_p$,$\sigma_p$) over the network output,

$$\hat{p}_i = Sigmoid(\hat{x}), \quad \hat{x}_i \sim \mathcal{N}(\mu_p, \sigma_p^2) \tag{4}$$

Since it is difficult to achieve an analytical solution for the expectation of loss in Eq. 3 with respect to $\hat{p}$, we adopt Monte Carlo integration to achieve an approximation. As a result, we learn the model with stochastic Binary Cross Entropy (BCE) loss for each pixel,

$$\mathcal{L}_{pixel} = \frac{1}{N} \sum_{i=1}^{N} \left( -z \cdot \log(\hat{p}_i) - (1 - z) \cdot \log(1 - \hat{p}_i) \right), \tag{5}$$

where $z$ is the groundtruth label, $\hat{p}_i$ is sampled from Eq. 4, and $N$ is the times of sampling. In conventional methods, noisy and outlying samples always cause high loss value for Eq. 3, and the optimization process tends to reduce the loss, thus those models are driven to learn uniformly from the noisy and outlying data. Yet in the probabilistic prediction model, the optimization process can drive the noisy samples to output a high variance of $\sigma_p$, thus allowing the model to account for noise and outliers. In such a way, the stochastic Binary Cross Entropy loss can be interpreted as learning to attenuate loss for uncertain samples with high variance estimation. To ensure numerically stable training, instead of directly predicting $\sigma_p$, we let the network output: $s = \log \sigma_p^2$. Then the $\sigma_p$ is computed via an exponential mapping: $\sigma_p = exp(\frac{s}{2})$, which always generates a positive value.

## 3.4  Image-level Learning with Bayesian Regression

In natural images, semantic classes are typically defined over image regions on a global scale rather than individual local pixels. In other words, a single pixel contains less information than a semantically-consistent region in images. Therefore, treating each pixel independently may lose global information, and lead to less effective models. In this part, we address this challenge by directly optimizing the image-level segmentation quality. Moreover, to learn more reliable visual-semantic mappings, we further formulate the image-level learning as a Bayesian regression problem, where the influence of noisy training samples is automatically explained away with a probabilistic distribution.

Instead of simply averaging the prediction accuracy over an image, we adopt the DICE Coefficient [13] as a quantitative measure for overall segmentation quality. Given a binary segmentation mask $P$ and the binary groundtruth mask $G$, the DICE Coefficient is computed as $\frac{2*|P \cap G|}{|P|+|G|}$, which equals to 1 when $P$ perfectly segment the mask as in $G$. To process with outputs by deep CNN based models, which are typically probabilities for each pixel belonging to a certain class, we utilize the soft DICE Coefficient in the form $\phi(P,G) = \frac{2\sum_{i=1}^{N} p_i g_i}{\sum_{j=1}^{N} p_j^2 + g_j^2}$, where $N$ is the total number of pixels, $p_i$ and $g_i$ are the segmentation probability and the groundtruth respectively for the $i$-th pixel. Since we aim to optimize the DICE Coefficient to be equal to 1, the objective for the overall segmentation quality can be formulated as a L1 regression problem as below,

$$\mathcal{L}_{dice} = |1 - \phi(P,G)| \tag{6}$$

where $\phi(P,G)$ is the aforementioned soft DICE Coefficient, $P$ and $G$ are the network's Sigmoid predictions and Groundtruth respectively. This object can also be formulated with L2 distance, which however is empirically found to be less effective.

Training the $\mathcal{L}_{dice}$ in Eq. 6 helps the model to learn more effectively visual-semantic mappings with global information. However, it is still vulnerable to noisy and outlying samples when training on seen classes. To address this, we propose to formulate the overall quality optimization as a problem of Bayesian regression, where estimated uncertainty helps to explicitly account for the observation noise in data. During training, we place over the soft-DICE output with a Laplacian distribution parameterized by $(\mu_i, \sigma_i)$, which are simultaneously estimated by the network as shown in the upper branch of Fig. 2. The $\sigma_i$ indicates the uncertainty that reflects the noise level of training samples; and $\mu_i = \phi(P,G)$ is the mean of DICE Coefficient computed with the predicted segmentation map $Sigmoid(\mu_p)$ and the groundtruth mask $G$. Consequently, we can derive a Bayesian regression loss which enables the model to learn to resist noisy samples [1],

$$\mathcal{L}_{img} = \frac{1}{\sigma_i}|1 - \phi(P,G)| + \log \sigma_i \tag{7}$$

As we can see, this objective can be seen as an adaptive loss attenuation, helping the model focus more on the representative samples while learning less from the noisy samples with high variance.

## 4  Experiments

### 4.1  Experimental details

**Datasets.** We adopt the two challenging benchmarks with large category sets and sufficient image samples for experiments, which are ADE20K [64] and Pascal-Context [42]. The ADE20K dataset contain 20K/2K/3K images for training/validation/testing respectively and provide a dense annotation of 150 categories including both objects and stuff. The Pascal-Context dataset consisting of both diverse indoor and outdoor images, which are split into 4998 training images and 5104 validation images. This dataset is annotated with more than 400 object and stuff classes, and the most frequent 59 classes are always adopted for benchmark evaluations.

**Settings and Evaluations.** Following previous work [2, 31], we experiment with varying numbers of unseen categories on these datasets. On ADE20K dataset, we randomly chose 25, 50, and 75 classes from the 150 categories set as unseen sets. On Pascal-Context dataset, we design two types of settings. One is based on the 59-class task, we vary the unseen class set by randomly choosing 10, 20,

| | K=10 | | K=20 | | K=30 | | | K=156 | | |
|---|---|---|---|---|---|---|---|---|---|---|
| | Overall | Unseen | Overall | Unseen | Overall | Unseen | | Unseen | IMN$^+$ | IMN$^-$ |
| DeVise | 44.5 | 14.0 | 36.0 | 7.8 | 30.3 | 6.5 | DeVise | 2.6 | 2.0 | 4.1 |
| GMMN | 45.5 | 25.5 | 36.6 | 20.0 | 31.6 | 15.2 | GMMN | 9.8 | 10.6 | 9.1 |
| CBN | 44.3 | 16.3 | 38.9 | 8.4 | 34.0 | 8.0 | CBN | 3.0 | 2.4 | 3.2 |
| Blank | 15.1 | 14.9 | 15.1 | 15.0 | 15.1 | 13.4 | Blank | 7.6 | 8.6 | 6.9 |
| Random | 9.2 | 9.1 | 9.3 | 9.0 | 10.0 | 8.3 | Random | 5.1 | 5.7 | 4.8 |
| *Baseline* | 47.7 | 27.9 | 39.2 | 18.3 | 33.2 | 14.9 | *Baseline* | 11.7 | 12.1 | 11.4 |
| *+U-Loss* | **48.1** | **35.4** | **42.0** | **24.5** | **36.5** | **18.6** | *+U-Loss* | **13.3** | **14.8** | **12.9** |

Table 1: Comparison (mIoU) on Pascal-Context dataset. ***Left***: performance for different number of unseen classes in the 59-classes setting. ***Right***: performance for the 156 unseen classes in the 215-classes setting. "IMN$^+$" represents the intersection between the 156 unseen classes and the Imagenet-1K class set. "IMN$^-$'' represents the rest classes in the 156 unseen classes. "Blank" means testing with all the pixels being foreground. "Random" means testing with pixels being randomly predicted as foreground with probability 0.15.

and 30 from the 59 classes as unseen classes, and take the rest of them as seen classes. In the other setting, we simulate the zero-shot semantic segmentation in practice by utilizing the 400+ densely labeled classes on Pascal-Context dataset. We take all the categories with at least two samples in the validation set, which results in 215 categories in total, where the most frequent 59 classes are used as seen classes, and the other 156 classes as the unseen set. In all the experiments, we report the mean Intersection over Union (mIoU) for performance evaluation.

**Baselines.** We formulate the zero-shot semantic segmentation as a pixel retrieval task. To achieve this, we concatenate visual features at each pixel and the semantic representation of a given class label, and apply several more layers to learn to compare the relation. The architecture of our baseline model is as in Fig. 2, except for the two uncertainty paths for $\sigma_i$ and $\sigma_p$.

**Implementation Details.** We apply semantic segmentation models like DeepLabV3+ [8] upon ResNet-50 [23] as fixed visual feature extractor, and adopt word embedding models like word2vec [40] to convert class tags into semantic representations. In our experiments, we fix the ResNet backbone with Imagenet [11] pretrained parameters, and randomly initializing all the other layers [22]. On both datasets, we set the weight $\lambda = 0.05$ for the loss in Eq. 2 and adopt Batchsize 8 and apply SGD [48] with learning rate $5 \times 10^{-4}$, momentum of 0.9, and weight decay $5 \times 10^{-4}$ to optimize the model for 20K iterations. Data augmentation including random horizontal flipping, random scaling (from 0.75 to 2), random cropping, and color jittering are applied in the training process. During testing, we input images at resolution $513 \times 513$, and threshold (with 0.5) the output to achieve binary output. We use Pytorch for model implementations and conduct all the experiments on a Titan Xp GPU.

## 4.2 Zero-shot semantic segmentation

In this section, we compare our method with previous methods to demonstrate the effectiveness of our framework. We summarize previous ZSS approaches and found that there are three types of methods that are highly related, which are metric learning-based DeVise [18, 55], variational mapping based GMMN [2, 31], and adaptive feature modulation based Conditional BatchNorm (CBN) [10]. For all these methods, we adopt DeepLabV3+ [8] for visual feature extraction and word2vec [40] for word embedding, and train with the same manner as ours. Moreover, since in our setting outputs are binary segmentation mask, we also evaluate with cases that constantly predicts all the pixels as foreground (denoted as *Blank*) or randomly predicts pixels as foreground (denoted as *Random*).

**Pascal-Context.** In the left part of Table 1, we vary the size of unseen classes (denoted as $K$) and compare the performance. As we can see, in all the settings, our model (denoted as "U-Loss") achieves much better overall and unseen-class performance than the existing methods. In particular, with 10 unseen categories, our method outperforms other methods by 10 percent. When testing with a larger "K", which means higher difficulty, our method still outperforms other methods by more than 3 percent. Compared to our baseline model (denoted as "Baseline"), the proposed uncertainty-aware learning achieves consistent and substantial accuracy improvement for overall classes and unseen classes, which validate the better generalization ability of visual-semantic mappings learned in our framework. In the middle rows of the table, we report performance for "Blank" and "Random".

| | K=25 | | K=50 | | K=75 | |
|---|---|---|---|---|---|---|
| | Overall | Unseen | Overall | Unseen | Overall | Unseen |
| DeVise | 27.7 | 7.9 | 22.8 | 5.8 | 19.1 | 5.9 |
| GMMN | 23.9 | 8.5 | 19.8 | 5.1 | 18.0 | 7.3 |
| CBN | 29.5 | 6.2 | 25.0 | 3.1 | 21.2 | 4.5 |
| *Baseline* | **32.0** | 12.7 | 27.4 | 10.1 | 24.7 | 13.5 |
| *+U-Loss* | 31.9 | **15.4** | **28.7** | **14.4** | **25.8** | **15.2** |

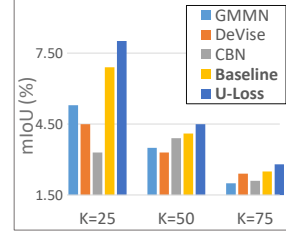

Table 2: Comparison (mIoU in%) on ADE20K validation set. **Left:** Performance with varying number of unseen classes. **Right:** Performance for generalized dense semantic segmentation. $K$ is the size of unseen classes. The mIoU of the unseen classes for Blank/Random settings is 11.2/ 5.8 (K=25), 9.3/ 5.1 (K=50), and 9.2/ 4.7 (K=75).

Compared to these two settings, our method performs much better with significant gaps, which demonstrates that our method does make meaningful predictions (Qualitative results are shown in Fig. 4 (a)). In the right part of Table 1, we show results for a more challenging setting, where the model is trained on 59 seen classes and tested with 156 unseen ones, as shown in the first column, our uncertainty based method achieves better performance than the baseline and previous methods. To further analyze our method, we separate the 156 unseen classes into two groups based on whether appearing in the 1K classes set of ImageNet[11] dataset or not. As shown in the last two columns, our network doesn't only improve the performance on the overlapped classes (denoted as "IMN$^{+}$"), but also boost performance for the "Real" zero-shot classes that are unavailable even in the backbone pretraining phase (denoted as "IMN$^{-}$").

**ADE20K.** Tab. 2 shows performance with varying number (from 25 to 75) of unseen classes on ADE20K dataset, which has 150 classes in total. Our model (denoted as "U-Loss") achieves the best mIoU for unseen classes in most of the settings in both overall and unseen classes. This again demonstrates the effectiveness of our method for learning reliable semantic-visual embeddings in zero-shot segmentation. To further evaluate the performance, we also consider the generalized evaluation which is challenging as it requires to make dense and disjoint segmentation with both seen and unseen classes. The results for unseen classes of different sizes are shown in the right of Tab. 2, as we can see, our method performs better than other methods.

### 4.3 Method Analysis

**Effects of different components.** At first, we analyze the effects of different components of our framework in Tab. 3. The first row corresponds to our baseline model without uncertainty-aware learning. As we can see, both the image-level uncertainty ($\mathcal{L}_{img}$) and the pixel-level uncertainty ($\mathcal{L}_{pixel}$) individually helps to improve accuracy for unseen classes in most cases. And when combining the two losses together for training, the performance can be further boosted. This demonstrates the effectiveness of our uncertainty-aware learning for reliable and generalizable visual-semantic mappings in zero-shot segmentation.

| $\mathcal{L}_{img}$ | $\mathcal{L}_{pixel}$ | PC-30 | PC-156 | ADE-75 |
|---|---|---|---|---|
| $\times$ | | 13.5 | 8.8 | 7.9 |
| | | 13.9 | 11.7 | 13.5 |
| | $\checkmark$ | 14.4 (+0.5) | 11.5 (-0.2) | 13.1 (-0.4) |
| $\checkmark$ | | 16.1 (+2.2) | 12.3 (+0.6) | 14.3 (+0.8) |
| $\checkmark$ | $\checkmark$ | 18.6 (+4.7) | 13.3 (+1.6) | 15.2 (+1.7) |

Table 3: Unseen classes perfromance (mIoU) with different training loss. "$\times$" indicates that the branch is removed from the model. "$\checkmark$" means that the uncertainty-aware loss is applied to the branch. "PC-30" and "PC-156" represents Pascal-Context with 30 and 156 unseen classes respectively. "ADE-75" is ADE20K dataset with 75 unseen classes.

**Different backbone.** We have shown that our method based on DeepLabV3+ [8] being visual feature extractor achieves better performance than other methods and the baseline. In this section, we will further validate the effectiveness of our method by adopting a different visual feature encoder, which is PSPNet [63]. In Fig. 3 (a), we utilized the ResNet50 initialized PSPNet and compare our uncertainty based method with the baseline. As we can see, our uncertainty based learning achieves much better performance on all the three challenging settings. This demonstrates our method's effectiveness and robustness to different visual feature extractors.

**Different word embedding.** In previous experiments, we adopted word2vec [40] for encoding semantic representations. In this part, we further validate our method by adopting the Fasttext [30]

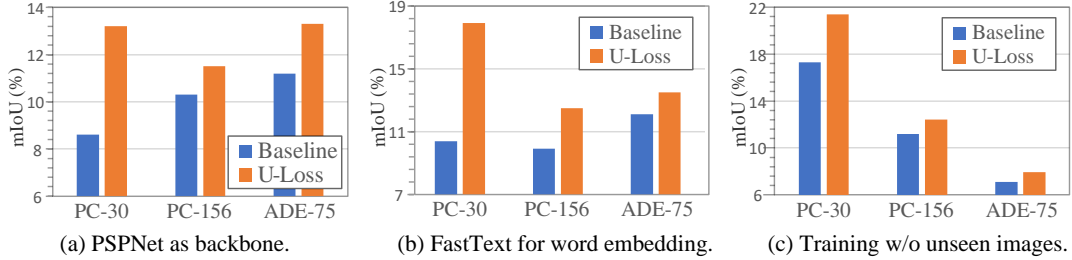

| (a) PSPNet as backbone. | (b) FastText for word embedding. | (c) Training w/o unseen images. |

Figure 3: Method analysis. "PC-30" and "PC-156" represents Pascal-Context with 30 and 156 unseen classes respectively. "ADE-75" is ADE20K dataset with 75 unseen classes.

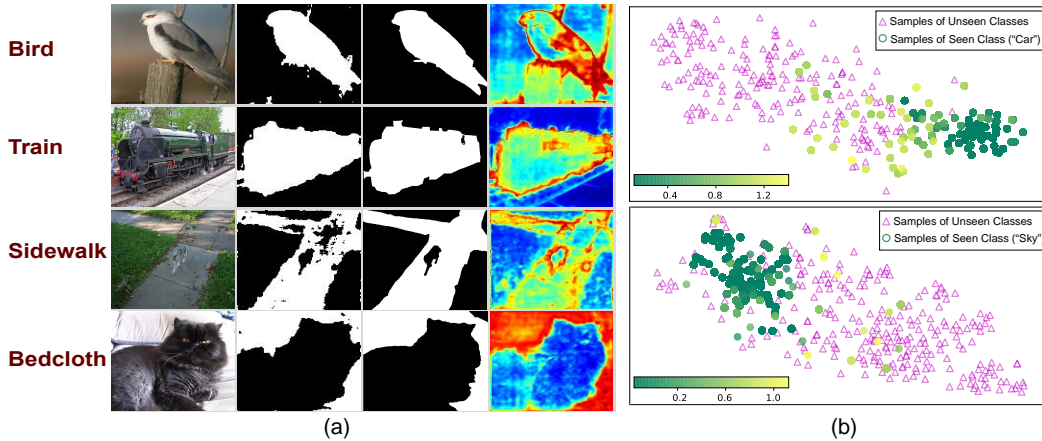

Figure 4: Qualitative results on PascalContext-60 dataset. (a) From left to right: unseen-class tag, input image, segmentation result, groundtruth, and pixel-wise uncertainty. (b) T-SNE visualizations of global uncertainty. Each data point indicates an image-level sample. The circles represent samples for a given seen class and the colors inside indicate the value (encoded by the colorbar) of the global uncertainty $\sigma_i$.

for word embedding. As shown in Fig. 3 (b), our uncertainty based learning can still achieve much better accuracy over the baseline models. This shows that our method is effective and robust for different types of language embedding models.

**Learn without unseen class in the background.** In semantic segmentation, each image is typically annotated with multiple classes. Therefore, unseen classes may be still processed by models, even though no supervisions are available. To exclude the influence of these unseen class samples existing in the background, we also train our models only with images containing no pixels belong to unseen classes. As reported in Fig. 3 (c), our uncertainty-based learning constantly improves the performance over the baseline. We surprisingly found that on Pascal-Context with 30 unseen classes, both the baseline and the final model achieves a better result than learning with unseen classes in the background. This may be because that excluding training images containing unseen classes leads to less training samples for seen classes, thus relive the over-fitting effect. Yet, in PC-156 and ADE-75, we find that too many samples are excluded, and consequently, the model performs worse.

**Analysis of uncertainty.** Finally, we provide an analysis of the uncertainty estimate in our model. We show in Fig. 4 (a) the estimated pixel-wise uncertainty map. As we can see, in most cases the boundary pixels show high uncertainty, as their feature are less discriminative compared to region center pixels. In Fig. 4 (b), we visualize the global uncertainty for a seen class against other unseen classes. As we can see, representative samples that are close to the class centers show low uncertainty (in darker colors). In contrast, noisy and outlying samples that are non-discriminative to unseen classes are estimated with higher uncertainty (in lighter colors), which means higher noise level and therefore are attenuated more in training.

# 5 Conclusion

In this paper, we propose a framework to learn reliable and robust visual-semantic mappings for zero-shot semantic segmentation. To resist the noise in training data, we leverage Bayesian uncertainty estimation to formulate the pixel-level and image-level accuracy prediction as a stochastic process, which naturally accounts for noisy and outlying training samples at both image level and pixel level. Therefore, the model learns attentively from representative samples, and suffers less from noisy samples. The effectiveness of our method is validated with extensive experiments. On multiple benchmarks, our method outperforms previous methods with a large gap.

## Acknowledgements.

This work was supported in part by DARPA and NSF Award No.1928477. The authors would like to thank the anonymous reviewers for constructive suggestions, as well as Jun Liu, Ximeng Sun, and Xueyan Zou for helpful discussions.

## Broader Impact.

Our work aims to learn reliable models for zero-shot semantic segmentation. From the technical perspective, our algorithm addresses the lack of training data in the open-set setting, thus benefiting a lot of real-world applications like image editing, open-world scene understanding. From the social responsibility perspective, our work aims for building machine learning algorithms with less training data, thus helping save both financial and energy costs during the data annotation process. The failures of our system may incur incorrect segmentation, which needs manual corrections from users before performing downstream tasks. The negative impact of our work could be that making companies, governments, or individuals more easily deploy these systems for unethical purposes.

## Footnotes

[1]See supplementary for more details.

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
