[Supplementary Material]

# Uncertainty-Aware Learning for Zero-Shot Semantic Segmentation —Supplementary Material—

## 1 Bayesian Regression for DICE Coefficient.

The L-1 regression problem can be interpreted as maximum likelihood with a Laplacian error model.

$$P(t_i|x_i, \mathcal{L}) = \frac{1}{2\sigma(x_i)} exp(-\frac{|t_i - y(x_i)|}{\sigma(x_i)}) \tag{1}$$

where $x_i$ is the input, $t_i$ is the target output, $y(x_i)$ and $\sigma(x_i)$ are the estimated mean and scale based on $x_i$ respectively. By taking negative log on Eq. 1, we have,

$$-\ln P(t_i|x_i, \mathcal{L}) = \frac{1}{\sigma(x_i)} \cdot |t - y(x_i)| + \ln(\sigma(x_i)) + \ln(2) \tag{2}$$

which needs to be minimized. The last term $\ln(2)$ is a constant, thus can be ignored. As a result we get the Bayesian regression loss with Laplacian error,

$$\mathcal{L} = \frac{1}{\sigma(x_i)} \cdot |t - y(x_i)| + \ln(\sigma(x_i)) \tag{3}$$

## 2 Effect of $\lambda$

| | PC-30 | | PC-156 | | ADE-75 | |
|---|---|---|---|---|---|---|
| | Overall | Unseen | Overall | Unseen | Overall | Unseen |
| $\lambda = 0.0$ | 36.0 | 17.1 | 23.1 | 12.1 | 25.3 | 13.9 |
| $\lambda = 0.05$ | 36.5 | 18.6 | 23.8 | 13.3 | 25.8 | 15.2 |
| $\lambda = 0.5$ | 36.5 | 17.9 | 23.9 | 13.6 | 25.6 | 14.5 |
| $\lambda = 1.0$ | 36.4 | 17.6 | 23.8 | 12.4 | 25.3 | 13.8 |

Table 1: Effect of $\lambda$ in Eq.(2) of the paper. As we can see, a too large or too small value for $\lambda$ decreases the unseen-class performance.