[Reviews · NeurIPS 2020]

Review 1

Summary and Contributions: The paper proposes to include in zero-shots semantic segmentation uncertainty estimation elements to model labeling and prediction uncertainties. The experimental results show that including these Bayesian uncertainty estimation elements provide noticeable gains in the obtained zero-shot segmentations.

Strengths: - The method description and the motivation behind it are reasonably well described. - The experimental design seems satisfactory, in particular when comparing with and without the uncertainty estimation elements. - The observed effect is surprisingly large, indicating that this is an important aspect to consider for the task.

Weaknesses: - To my understanding, the experimental section only compares results generated for this paper. This is good because it keeps apples-to-apples comparisons, however it is suspicious since the task is not novel. Some comparison with results from other works (or a justification of why this is not possible/suitable) would be welcome. For example [2, table 3] seems to have directly comparable results, yet these are nowhere mentioned in this paper. - Albeit the observed effects are strong, it remains unclear “why does the method work?” in particular regarding the L_pixel component. Providing stronger arguments or intuitions of why these particular losses are “bound to help” would be welcome.

Correctness: - Unclear how the parameters of section 4.1.4 were selected. One should assume these are thus “best-fit results over test-set”. - The final line L297 claims “our method outperforms previous methods”, yet the paper provides zero quantitative comparison to numbers reported in previous papers. Thus this final claim seems unsupported. (same in L261). Other than that the paper seems correct.

Clarity: Yes. Other than the minor comments in “additional feedback”, overall the text is clear and well written.

Relation to Prior Work: Section 2 on related work is satisfactory, some suggestions for improvement: - I would suggest adding a discussion of “calibrated probabilities” literature as one more way to obtain good uncertainty estimates. I am thinking of works like “Well-calibrated Model Uncertainty with Temperature Scaling for Dropout Variational Inference”, Neurips workshops 2019; and related ideas. - Additionally I would suggest a mention/bridge to the “pixel-level grounding” literature. Albeit the evaluation datasets are a bit different, the key methodologies are similar. I am thinking of works like “Segmentation from Natural Language Expressions”, ECCV 2016; and related papers in that space.

Reproducibility: Yes

Additional Feedback: Here some detailed comments and questions: Fig 1a: is that label correct for 1a ? Seems correct for 1b and 1c. L52: At this point nothing seems segmentation specific. Does this method also work for zero-shot classification ? Maybe hint at what is pixel-level specific. L118: why would atypical samples be considered noise ? Being atypical does not make it “wrong” or “noisy”. I am missing a logical link. Maybe the explanation needs to be refined. L125: is there a reason to keep the “post-concatenation with semantic vector head” so shallow ? 1x1 conv + sigmoid, could be instead multiple layers, could it not ? L141: It is not clear to me the process that drives sigma_p up. Similarly L142 does not follow for me. Please add citation or expand reasoning. L155: Maybe mention why DICE and not IoU itself ? L156: Would be best for understanding to explain earlier what is meant by “image-level”, since the DICE coefficient is computed by going over the pixels still. L162: For the sake of pedagogy make explicit the link between Laplacian errors and the L1 loss is (6). L174: My intuition for sigma_i is that this is a prediction of the “class-image” confidence (“does the network recognize this as a typical image for class X?”). Do you share this intuition ? Maybe worth mentioning. Eq 7: for sake of pedagogy would be good to explain/justify the terms added. I can make heads and tails of it, but probably best to remove the guesswork for the reader. Also, did you try other formulations for this loss ? Sec 3: Possibly good to mention that an ablation study of the losses is present in section 4.3. Table 1: Devise results are quite bad, would be good to mention why. Also mentioning why is the baseline so much better than the other methods considered. As mentioned in “weaknesses” including some numbers of other papers would be good too, for context (either here, in text, or in a new table). L204: Do I understand correctly that the backbone is frozen ? Is this to save computing cost ? To avoid overfitting to seen categories ? A brief justification is welcome. L205: Why is lambda so small ? Should it not be closed to 1 ? How was it selected ? Table 2: Adding Random and Blank baselines welcome. Otherwise it looks as if there is something to hide. Fig 3b: what is the intuition behind the improved PC-30 results ? L260: hav e L261: “significantly better than previous methods” -> not really since no inclusion of results reported in other papers nor mention/discussion of how your implementations compared to these. Either rephrase or add more support to this claim. L277: bas eline. Fig 4b: what are the axes here ?


Review 2

Summary and Contributions: This paper proposes an uncertainty aware zero-shot semantic segmentation technique. At the pixel level visual feature maps and semantic representation vectors are concatenated and 1x1 conv. Layers are learned over concatenated features. The standard binary cross entropy (BCE) loss is replaced with the stochastic BCE loss where an uncertainty parameter is estimated to quantify the noise level of the data. Instead of directly using the logit output for a given pixel a Gaussian distribution centered at this logit output with a variance is defined and sigmoid probability are derived based on a sample drawn from this distribution. The higher the estimated variance of this distribution is the lower the impact of the pixel will be on the loss function. At the image level a loss function based on Dice coefficient is used to ensure globally coherent visual-semantic mapping. To account for observation and label noise in seen classes the optimization is formulated as a Bayesian regression problem. ResNet pretrained with Imagenet is used as a backbone for visual feature extractor. ADE20K and Pascal-Context datasets are used for evaluation. Varying numbers of unseen categories are considered in experiments. The proposed approach is compared against three techniques. A metric learning based technique DeVise, variational mapping based technique GMMN, adaptive feature modulation based CBN. All techniques use DeepLabV3+ for visual feature extraction and word2vec for semantic embedding. Additionally a variant of the proposed technique that removes two uncertainty paths is considered as a baseline. Results suggest that the proposed approach achieves better overall and unseen-class performance than the existing methods including the baseline approach. The effects of image and pixel-level uncertainty paths are independently analyzed. Results show that adding each of this component improves performance with the most performance improvement achieved when both components are included. The effect of visual feature encoder is investigated by replacing DeepLabV3+ by PSPnet feature encoder. Results suggest that the proposed approach continue to outperform the baseline approach when PSPnet is used as a feature encoder. Similarly word2vec is replaced by Fasttext to investigate the effect of word embedding. Results show the improvement achieved over baseline is still preserved with a different word embedding technique. Most of the seen classes in images coexist with unseen classes. To eliminate the effect of using images of unseen classes during training additional experiments that only use images with seen classes are performed. The improvement over the baseline can still be maintained. Finally, some examples of pixelwise uncertainty heat maps are provided to show that pixels with the largest uncertainty are usually those that exist in the borders.

Strengths: 1. Modeling uncertainty that emerge in the form of label and observation noise in seen class training samples to attenuate the effect of samples with higher uncertainty in a zero-shot semantic segmentation framework 2. Use of global information learned from all images to achieve spatially coherent segmentation, an aspect of segmentation that cannot be handled by independent pixel-level classification 3. Experiments rigorously cover various scenarios that show the proposed approach is not backbone dependent.

Weaknesses: 1. It is not clear how this model distinguishes between label and observation noise. 2. Some more recent work is not included in empirical comparison

Correctness: Yes the claims and methods are correct to the extent they can be verified.

Clarity: The paper is well written and easy to read.

Relation to Prior Work: Prior work seem to be well covered.

Reproducibility: No

Additional Feedback: It is not clear how this model distinguishes between label and observation noise. Pixel level noise is modeled by a Gaussian distribution and image-level noise is modeled by a Laplacian distribution. Does this framework assume observation noise occurs at pixel level and label noise occur at the image level? As a baseline zero-shot semantic segmentation is considered as a pixel retrieval task that uses the same architecture as the proposed approach. The only differences being the two paths involving noise variances. The main motivation for such a choice is to show that modeling noise in seen classes improves the performance, which is perfectly fine. However, given that one of the main motivations of the paper is to show that semantic segmentation that uses structural information will be better than that treat pixels independently another baseline that classifies each pixel independently would be very useful. The three benchmarks adopted do not seem to fill this empirical gap. This recent work from the literature (Bucher et al, NIPS 2019) though cited is not included as a benchmark for the experiments. As a generator-based technique it could have been considered under a new category. Minor: “Effective of different components” Did the authors mean “Effects of different components” Author responses address my confusions about pixel vs. image noise as well as the benchmark not including any generative technique. I am increasing my rating from 6 to 7 now.


Review 3

Summary and Contributions: This paper proposes a novel method for zero-shot semantic segmentation. It introduces an uncertainty estimation module to improve the robustness of noise.

Strengths: This paper is the first to consider the effect of noise in zero-shot semantic segmentation and propose an uncertainty-aware solution. From ablation experiments, it shows the proposed module can improve the results.

Weaknesses: In L282 Analysis of uncertainty, it is better to provide the mIoU on different subsets on different uncertainty levels. It will demonstrate that uncertainty/noise will affect the results. From the experiments section, the improvements brought from the uncertainty estimation module are marginal.

Correctness: Yes

Clarity: Yes

Relation to Prior Work: I am unfamiliar with zero-shot learning.

Reproducibility: Yes

Additional Feedback:

[Author Response · NeurIPS 2020]

We thank all the reviewers for their valuable comments and positive feedback: R1) Indicating an important aspect of the zero-shot task; R3) Rigorous experiments; R4) A novel method. All the reviewers are satisfied with our paper's contributions, good performance, and clear presentation. Codes for reproducing all the experiments will be released.

**Response to Reviewer-1.**

*R1.1 Comparisons with other papers.* We note that previous works [2,27,48] experiment with different settings and datasets, which makes it hard to conduct unified and direct comparisons. We also considered ZS3Net[2] for direct comparison, yet found that its setting may be different from ours. We assume no knowledge of unseen classes in training, while ZS3Net learns implicitly with supervision of unseen classes. We would like to refer reviewers to Issue #1, #4, #6 raised (by other researchers) in the official Github repo of ZS3Net for contrasting the settings. To ensure fair comparisons in our work, for the GMMN we adopted the implementation from ZS3Net. And all the methods are trained similarly with independently selected hyper-parameters for each.

*R1.2 How our method works.* As motivated in Fig.1, in zero-shot learning, optimizing over the noisy/abnormal samples of seen classes may result in models with a biased visual-semantic mapping, thus the inferred classier for unseen classes may be less reliable. In the scenarios of semantic segmentation, the noise appears in two levels: one is at image-level, e.g. some van cars look like 'bus' but are labelled as 'car'; the other is at pixel-level, e.g. pixels near boundaries are hard to distinguish and easily mis-annotated. Our uncertainty-aware model is proposed to address such challenges. We observed that the image-level learning path is critical to the accuracy of segmentation; yet, at the same time, the pixel-level learning path plays as a complementary role in learning refined segmentation details.

*R1.3 How the uncertainty-aware learning works.* Due to the noisy data collection and annotation process in practice, data samples may have different levels of uncertainty. For example, an abnormal sample is expected to make a less confident prediction than a typical sample, and pixels in the object centers are expected to be more confident than boundary pixels. The level of noise introduced for individual samples is called Heteroscedastic uncertainty and can be formulated as a random variable and estimated nonparametrically along with the learning of deep neural networks [12, 19, 28, 32, 38]. Specifically, in deep networks for classification or regression, the model is trained to learn a feature mapping such that the features are distributed in the form of a certain exponential family, with a unique variance and mean for all samples. Optimizing the feature mapping over noisy/abnormal samples may result in a biased distribution. Yet, explicitly modeling the uncertainty $\sigma$ for each sample, as in our formulation, allows the model to adaptively adjust the variance for individual samples. As a result, all the data samples can be properly accounted for by the model with less bias toward the abnormal/noisy samples. During the optimization process, the abnormal/noisy samples are typically mapped far from the majority of the samples, thus leading to high uncertainty. We will clarify this in the paper.

*R1.4 Parameter selection.* We select hyper-parameters based on a validation set split from the seen-class training set.

*R1.5 Eq 7: for sake of...* We also tried with L-2 loss, but found the accuracy drops in all settings.

*R1.6 Table 1: DeViSE results...* One main reason may be that DeViSE maps the visual feature into the low-dimensional semantic space for nearest neighbor search. Such a process may shrink useful information and aggravates the Hubness problem in zero-shot learning. Moreover, in this work, we test with a much larger unseen class set than in previous works, thus further increasing the difficulty of classification for DeViSE.

*R1.7 Table 2: Adding Random...* We removed these two lines from this table to align the height of the figure in the right. On ADE20k data, the mIoU (in %) of the unseen classes for Blank/Random is 11.2/ 5.8 (K=25), 9.3/ 5.1 (K=50), and 9.2/ 4.7 (K=75), which is much lower than our final accuracy. We will add this back.

*R1.8 Fig 3b: what is the intuition...* We guess that R1 refers to Fig 3c. We found this is a dataset-specific phenomenon and observed consistent improvements of unseen-class performance for all the methods on PC30 under this setting. The reason may be that removing the unseen-class images results in smaller but more compact training set for seen classes, thus leading to better visual-semantic mappings for inferring model for unseen classes. We also observed that the overall mIoU for our final model drops from 36.5% to 28.6% when removing all the unseen-class images for PC30.

*R1.9 Other comments.* We appreciate the detailed and very helpful comments. Due to the page-limit, we can only address part of them in this rebuttal. We will carefully address all the comments in a revised version of our paper.

**Response to Reviewer-3.**

*R3.1 About pixel and observation noise.* Yes, noise exists at both the image- and pixel-level. Please see our response to R1.2 and R1.3 for details.

*R3.2 Baseline classifying pixels independently.* Thanks for the suggestions. With only the pixel-level path, the model achieves mIoU for unseen classes with 13.5% for PC30, 8.8% for PC156, and 7.9% for ADE75. Compared to the results in Tab.3, we found that simply adding the image-level path can improve about 0.4%, 1.9%, and 5.6% for PC30, PC156, and ADE75 respectively. We will update our paper.

*R3.3 Generator-based techniques.* We denote [2] as GMMN in our experiments. Please also see our response to R1.1.

**Response to Reviewer-4.**

*R4.1 Results for different uncertainty/noise level.* Thanks for the suggestions. To evaluate the effects of different levels of noise, we randomly shuffle the class labels for a training sample. The unseen-class mIoU (in %) for baseline/U-loss under different probability on PC-156 are 11.7/13.3 (p=0.0), 11.1/12.8 (p=0.1), 10.9/12.9 (p=0.25), 9.6/12.1 (p=0.5), which shows that uncertainty-aware learning achieves better robustness. We will update the paper.

[Meta-Review · NeurIPS 2020]

This paper presents an interesting two-branch framework to address the zero-shot semantic segmentation problem. The approach is one of the first to utilize uncertainty modeling, both at the pixel and image level to model label/observation noise, in the zero-shot setting. The reviewers appreciated the approach and the results, but expressed some concerns about clarity of the method (esp. with regard to addressing both label and observation noise) and comparisons to other works. The rebuttal addressed some of these concerns, and the clarifications should be added to the camera-ready version. Overall, this paper has a nice contribution to the sub-field that would be of interest to the community.